# On Exotic Objects Made of Dark Energy and Dark Matter: Mass-to-Radius Profiles and Tidal Love Numbers

**Camila Sepúlveda and Grigoris Panotopoulos \***

Departamento de Ciencias Físicas, Universidad de la Frontera, Casilla 54-D, Temuco 4811186, Chile; camila.sepulveda.rivas@gmail.com

\* Correspondence: grigorios.panotopoulos@ufrontera.cl

**Abstract:** We investigate some properties of exotic spherical configurations made of dark matter and dark energy. For the former, we adopt a polytropic equation-of-state, while for the latter, we adopt the extended Chaplygin gas equation-of-state. Solving the Tolman–Oppenheimer–Volkoff equations, within the two-fluid formalism, we compute the factor of compactness, the mass-to-radius relationships, as well as the tidal Love numbers and dimensionless deformabilities. A comparison between single-fluid objects and two-fluid configurations is made as well.

**Keywords:** relativistic stars; equations of state; composition of astronomical objects; binaries

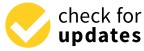



## 1. Introduction

According to the $\Lambda$CDM cosmological model, the Universe is not only composed of ordinary matter, but also of a strange substance known as dark matter (DM) [1]. The $\Lambda$CDM model, based on cold dark matter and a positive cosmological constant, has been very successful as it is in excellent agreement with observational data. If we study massive stellar objects, such as galaxies or clusters of galaxies, it can be inferred that there is more matter than the visible one. The existence of dark matter was speculated for the first time when Fritz Zwicky studied the Coma cluster of galaxies, and determined that its mean density was 400 times greater than that derived from observations of light matter [2]. Later on, Rubin and Ford obtained the rotation curve of the Andromeda galaxy, and they observed that the speed along all the arms of the galaxy remained almost constant, which suggested the presence of matter in the halo that we cannot see [3]. For a discussion on current evidence for dark matter or galaxy rotation curves, see for instance [4–7].

Dark and ordinary matter are not the only components that exist in the Universe according to the $\Lambda$CDM model [8], since it proposes the existence of an energy that currently accelerates the expansion of the Universe, and that represents around 70% of the energy content of the Universe ($\sim$25% dark matter and $\sim$5% ordinary matter), known as dark energy (DE). For some more recent cosmological constraints coming from supernova data, see, e.g., [9–12], and also see [13,14] for historical research.

Compact objects [15], such as neutron stars [16] and strange quark stars [17–22] (provided that the latter exist in Nature), are the final fate of sufficiently massive stars, since less massive stars lead to white dwarfs, which are considerably less relativistic. Those objects are characterized by ultra-high matter densities and strong gravitational fields. As matter becomes relativistic under those extreme conditions, the Newtonian description is inadequate, and therefore a relativistic description is more appropriate. In addition to that, there are studies of less conventional or even exotic stars. It has been proposed that stars made entirely of self-interacting dark matter may exist, called boson stars [23–28] (see, e.g., [29,30] for recent progress in the numerical simulations of boson star binaries). See [31] for a classic review of boson stars and [32] for a very recent one.

The properties of compact stars made of ordinary matter admixed with condensed dark matter or fermionic dark matter were also extensively studied, as can be seen, for

instance, in [33–43]. Similarly, there are a few works where the authors contemplate the possibility that exotic objects made of dark energy might exist as well [44–47].

Despite the success of the ΛCDM model at large (cosmological) scales (see, however, [48,49] for challenges), one may go beyond the concordance model for several reasons. First, as far as DM is concerned, beyond the collisionless dark matter paradigm, the self-interacting DM [50] has been proposed as an attractive solution to the dark matter crisis at short (galactic) scales, for reviews see, e.g., [51,52]. Similarly, given that current cosmic acceleration calls for dark energy, in some recent works, the authors proposed to study non-rotating dark energy stars [44–47] assuming an extended equation-of-state (EoS) of the form $p = -B^2/\rho^\alpha + A^2\rho$ [53,54], with $A$, $B$ and $\alpha$ being constant parameters, where $p, \rho$ are the pressure and the density of the fluid, respectively. A simplified version of this, namely $p = -A/\rho^\alpha$, with $0 < \alpha \leq 1$, known as generalized Chaplygin gas equation-of-state, was introduced in cosmology a long time ago to unify the description of non-relativistic matter and the cosmological constant [55,56].

In a binary system, one of the stars is subject to the gravitational field produced by its companion. Due to the gravitational interaction, tidal forces are generated. The theory for tidal deformability was introduced by Love [57,58] more than 110 years ago to study the deformability of the Earth due to the tides produced by the Moon. Love introduced dimensionless numbers, the tidal love numbers, such as $k$, which describes the relative deformation of the gravitational potential. Those numbers can also be used for the studies of relativistic stars in binary systems [59–64]. In the case of two compact objects, each star produces deformabilities, or tides on its surface, towards the other, due to the mutual gravitational attraction. Those deformabilities may be quantified introducing $\Lambda$, which is a dimensionless number that can be calculated in terms of the tidal Love number $k$, as can be seen in the discussion below.

Through observations from gravitational waves (GWs) from mergers of binaries containing neutron stars and/or black holes, it has been possible to obtain information about the nature of the colliding bodies, thanks to the GW detectors LIGO and Virgo [65–67], see [68] for their latest catalogue paper. KAGRA [69] and LIGO-India [70] will also contribute to such future observations, and future ground-based GW detectors (Einstein Telescope [71] and Cosmic Explorer [72]) are planned for the future. Low-frequency space-based GW detectors, such as LISA [73,74], have also been planned, although space-based detectors are not expected to constrain the neutron star EoS, and LISA will not be able to probe the tidal deformability of objects with the masses we are considering here. For constraints on the nuclear equation-of-state that may be obtained from GW observations, see, for instance, [75–78], and for constraints on boson stars from putative binary black hole signals, see, e.g., [79,80].

In this article, we propose to investigate some properties of spherical configurations made of both dark matter and dark energy. To be more precise, the goal here is two-fold, namely generate the mass-to-radius profiles of isolated objects, as well as compute the tidal Love numbers and deformabilities in binaries. It would be interesting to see what can be expected if those objects do exist in Nature, and also to know how their properties differ as compared to other classes of compact objects.

Our work is organized as follows: after this introduction, we present the structure equations governing hydrostatic equilibrium as well as the tidal Love numbers in Section 2. The equations-of-state employed here and our numerical results are shown and discussed in Section 3. Finally, we summarize and conclude our work in Section 4. We adopt the mostly positive metric signature $(-,+,+,+)$, and we work in natural geometrical units setting $c = 1 = G$.

## 2. Theoretical Framework: Notation and Setup

### 2.1. Hydrostatic Equilibrium

We work with relativistic configurations, in four dimensions, with a vanishing cosmological constant, and assuming that the objects are electrically neutral and non-rotating.

The metric for static, spherically symmetric space-times in Schwarzschild-like coordinates, $(t, r, \theta, \phi)$, is given by

$$ds^2 = -e^\nu dt^2 + e^\lambda dr^2 + r^2(d\theta^2 + \sin^2\theta\, d\phi^2), \tag{1}$$

for interior solutions, $0 \leq r \leq R$, where $R$ is the radius of the star, while for the discussion to follow, it is more convenient to work with the mass function, $m(r)$, defined by

$$e^\lambda = \frac{1}{1 - \frac{2m}{r}}. \tag{2}$$

To obtain interior solutions describing the hydrostatic equilibrium of dark stars, one needs to integrate the Tolman–Oppenheimer–Volkoff equations [81,82], for a two-fluid spherical configuration, i.e., made of both dark matter and dark energy, given by the equations in the two-fluid formalism [83,84]

$$m'(r) = 4\pi r^2 \rho(r), \tag{3}$$

$$p'_M(r) = -[\rho_M(r) + p_M(r)]\, \frac{m(r) + 4\pi r^3 p(r)}{r^2(1 - 2m(r)/r)}, \tag{4}$$

$$p'_E(r) = -[\rho_E(r) + p_E(r)]\, \frac{m(r) + 4\pi r^3 p(r)}{r^2(1 - 2m(r)/r)}, \tag{5}$$

$$\nu'(r) = 2\, \frac{m(r) + 4\pi r^3 p(r)}{r^2(1 - 2m(r)/r)}, \tag{6}$$

where a prime denotes differentiation with respect to $r$, and where $p_M$ and $\rho_M$ are the pressure and the energy density for dark matter, respectively, while $p_E$ and $\rho_E$ are the corresponding quantities for dark energy, respectively. The total pressure and density are, respectively,

$$p = p_M + p_E, \tag{7}$$

$$\rho = \rho_M + \rho_E. \tag{8}$$

The function $\nu(r)$ may be computed by

$$\nu(r) = \ln\left(1 - \frac{2M}{R}\right) + 2\int_R^r \frac{m(x) + 4\pi x^3 p(x)}{x^2(1 - 2m(x)/x)}\, dx, \tag{9}$$

with $M$ being the mass of the star. The Equations (3)–(5) are to be integrated, imposing at the center of the star appropriate conditions

$$m(0) = 0, \tag{10}$$

$$p_M(0) = p_{cM}, \tag{11}$$

$$p_E(0) = p_{cE}, \tag{12}$$

where $p_{cM}$ is the central pressure for dark matter and $p_{cE}$ for dark energy. In addition, the following matching conditions must be satisfied at the surface of the object

$$p(R) = 0, \tag{13}$$

$$m(R) = M, \tag{14}$$

taking into account that the exterior vacuum solution is given by the Schwarzschild geometry.

*2.2. Tidal Love Numbers*

In a binary system, each star is subjected to the gravitational field produced by its companion. This interaction produces tides on the surface of stars.

For a spherical body, we used a dimensionless number, $k$, that describes the relative deformation of the gravitational potential. The tidal Love number, $k$, was originally introduced by Love [57,58], while the relativistic theory of tidal Love numbers was formulated in [59–64]. In the latter case, $k$ is computed by

$$k = \frac{8C^5}{5} \frac{K_o}{3K_o \ln(1 - 2C) + P_5(C)}, \tag{15}$$

where $C = M/R$ is the factor of compactness of the star, and

$$K_o = (1 - 2C)^2 \left[ 2C(y_R - 1) - y_R + 2 \right], \tag{16}$$

$$y_R \equiv y(r = R). \tag{17}$$

$P_5(C)$ is a fifth-order polynomial given by

$$P_5(C) = 2C \left[ 4C^4(y_R + 1) + 2C^3(3y_R - 2) + 2C^2(13 - 11y_R) + 3C(5y_R - 8) - 3y_R + 6 \right]. \tag{18}$$

While the function $y(r)$ satisfies a Riccati differential Equation [64]:

$$ry'(r) + y(r)^2 + y(r)e^{\lambda(r)}[1 + 4\pi r^2(p(r) - \rho(r))] + r^2 Q(r) = 0, \tag{19}$$

supplemented by the initial condition at the center, $y(0) = 2$, where

$$Q(r) = 4\pi e^{\lambda(r)} \left[ 5\rho(r) + 9p(r) + \frac{\rho(r) + p(r)}{c_s^2(r)} \right] - 6\frac{e^{\lambda(r)}}{r^2} - [\nu'(r)]^2, \tag{20}$$

and with $c_s^2 \equiv dp/d\rho = p'(r)/\rho'(r)$ being the speed of sound. The speed of sound for each fluid component may be computed using either the solution $p_i(r)$, $\rho_i(r)$, with $i = M, E$, or the definition since the EoS is known. However, for the total pressure and energy density, as there is no EoS for those, the speed of sound is computed once the solution is obtained.

It has been pointed out in [61,63,64] that, when phase transitions or density discontinuities are present, it is necessary to slightly modify the expressions above. Since, in the present work, the energy density takes a non-vanishing surface value, $\rho_s$, we incorporated the following correction in our analysis:

$$y_R \rightarrow y_R - 3\frac{\Delta\rho}{\langle\rho\rangle}, \tag{21}$$

where $\Delta\rho = \rho_s$ is the density discontinuity, and

$$\langle\rho\rangle = \frac{3M}{4\pi R^3}, \tag{22}$$

is the mean energy density throughout the object.

In terms of the tidal Love number, we may determine the deformabilities of the star produced by the gravitational field of its companion. These are determined by

$$\lambda \equiv \frac{2}{3}kR^5, \tag{23}$$

$$\Lambda \equiv \frac{2}{3}\frac{k}{C^5}, \tag{24}$$

where $\lambda$ is the tidal deformability and $\Lambda$ is the dimensionless deformability, derived from the relationship between the induced quadrupole moment $Q$ and the applied tidal field $\epsilon$

$$Q_{ij} = -\lambda\,\epsilon_{ij} \equiv -\frac{2}{3}kR^5\,\epsilon_{ij}. \tag{25}$$

The interested reader may consult [59–64] for more details.

The Riccati Equations (19) and (20) correspond to a single-fluid object. In the two-fluid formalism, the correct equation is obtained, making the substitution [85]

$$\frac{\rho(r) + p(r)}{c_s^2(r)} \rightarrow \frac{\rho_M(r) + p_M(r)}{(p_M'(r)/\rho_M'(r))} + \frac{\rho_E(r) + p_E(r)}{(p_E'(r)/\rho_E'(r))}. \tag{26}$$

However, in our work, we numerically verified that Equations (20) and (25) lead to the same results for the tidal Love numbers and deformabilities. This is no surprise, since using the structure equations and the definition of the sound speed, both sides of the above equation give $-2\rho'(r)/\nu'(r)$.

Finally, it is worth mentioning that, due to the overall factor $(1 - 2C)^2$ in the expression for $k$, clearly the tidal Love number for black holes is precisely zero as $C = 1/2$, or at least for the Schwarzschild black hole of general relativity [62]. On the contrary, for any other relativistic object (neutron stars, quark stars or dark stars) $C < 1/2$ and $k \neq 0$. Moreover, since the dark sector does not directly interact with light, one expects only gravity waves emitted from dark binaries without the electromagnetic counterpart, similarly to black hole binaries. Those properties may allow us to differentiate between different classes of objects.

## 3. Equation-of-State and Numerical Results

### 3.1. Equations-of-State

To integrate the TOV equations, we must specify the equation-of-state of the matter content. In the two-fluid formalism, since there is no direct interaction between the two fluid components, each fluid is characterized by its own EoS.

Recently, there has been a reinterpretation of DM as a Bose–Einstein Condensate [86–88] as it solves the cusp/core problem [89]. The single wave-function describing the Bose–Einstein condensation [90–92] satisfies the Gross–Pitaevskii Equation [93,94], also known as non-linear Schrödinger equation, as can be seen in, e.g., [95–98] (see also [99–103] for the cosmological implications of ultra-light bosons). The presence of the non-linear term, interpreted as enthalpy, gives rise to a polytropic EoS of the form [95–97]

$$p(r) = K\rho^2(r), \tag{27}$$

where the constant of proportionality, $K$, is computed to be [35,95,96]

$$K = \frac{2\pi l}{m^3}, \tag{28}$$

with $m$ being the mass of the dark matter particle, and $l$ being the scattering length that determines the cross-section of 2 by 2 elastic scattering processes, $\sigma = 4\pi l^2$ [35,95,96]. Assuming numerical values for the free parameters of the model, $m, l$, as follows: $m \sim 1$ GeV and $l \sim 10$ fm [104], then the constant $K$ is found to be $K = 62.83$ fm/GeV$^3$ = 318.4 GeV$^{-4}$.

Then, the EoS for dark energy is assumed to be the extended Chaplygin gas equation-of-state [53,54]

$$p(r) = -\frac{B^2}{\rho(r)} + A^2\rho(r), \tag{29}$$

where $A$ and $B$ are positive constants, out of which $A$ is dimensionless, while $B$ has dimensions of energy density and pressure. Also, note that Chaplygin's EoS is given by $p = -B^2/\rho$ [55], while the additional barotropic term corresponds to $A^2\rho$, also leading

to a good DE model [105]. The numerical values of $A$ and $B$ considered here are shown in Table 1.

**Table 1.** Numerical values of $A$ and $B$ for the three different DE models [46,47].

|  | **A** | **B** |
|---|---|---|
| Model 1 | $\sqrt{0.4}$ | $0.23 \times 10^{-3}/\text{km}^2$ |
| Model 2 | $\sqrt{0.425}$ | $0.215 \times 10^{-3}/\text{km}^2$ |
| Model 3 | $\sqrt{0.45}$ | $0.2 \times 10^{-3}/\text{km}^2$ |

The choice of the numerical values of the parameters has been made with two criteria in mind: first, to obtain realistic, well behaved solutions, and in addition to that, to model objects with properties similar to the ones of neutron stars, namely radii of the order of 10 km as well as stellar masses of the order of solar mass. We should mention, however, that according to our numerical results, the highest masses we obtained here are notably lower than the masses of the most massive pulsars, as can be seen in the discussion below in Section 3.2. It must be taken into account that, given the form of the EoS, the pressure and density cannot become zero at the same time. Therefore, a vanishing pressure at the surface of the star implies a non-vanishing value for the energy density, which is found to be $\rho_s = B/A$.

Furthermore, regarding the conditions at the center of the objects, for the numerical analysis, it turns out that it is convenient to introduce a new dimensionless factor defined as

$$f = \frac{p_{cM}}{p_{cM} + p_{cE}},\tag{30}$$

and, in the following, we shall consider three different values, namely $f = 0.6, 0.8, 0.95$.

### 3.2. Discussion of the Results

Our numerical results are displayed in Figures 1–6 below, where we vary both the factor $f$ and the pair of constants $A, B$ related to the dark energy model (Table 1). The factor of compactness, $C$, the tidal Love number, $k$, as well as the deformability, $\Lambda$, are dimensionless quantities, whereas the stellar mass, $M$, as well as the stellar radius, $R$, are dimensional quantities, the former in solar masses and the latter in km.

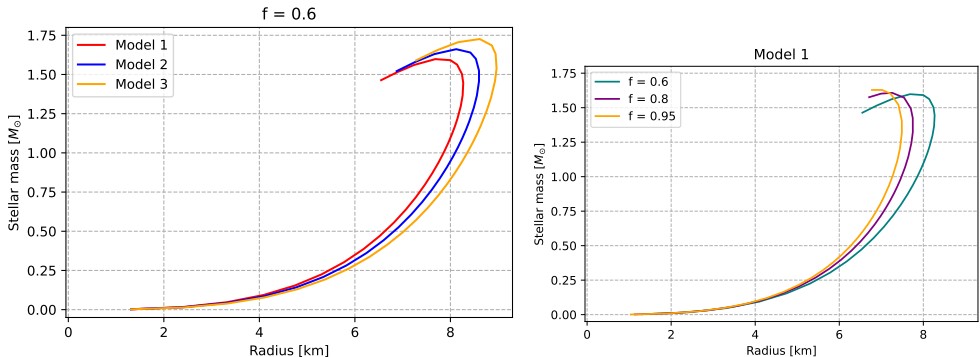

**Figure 1.** *Cont.*

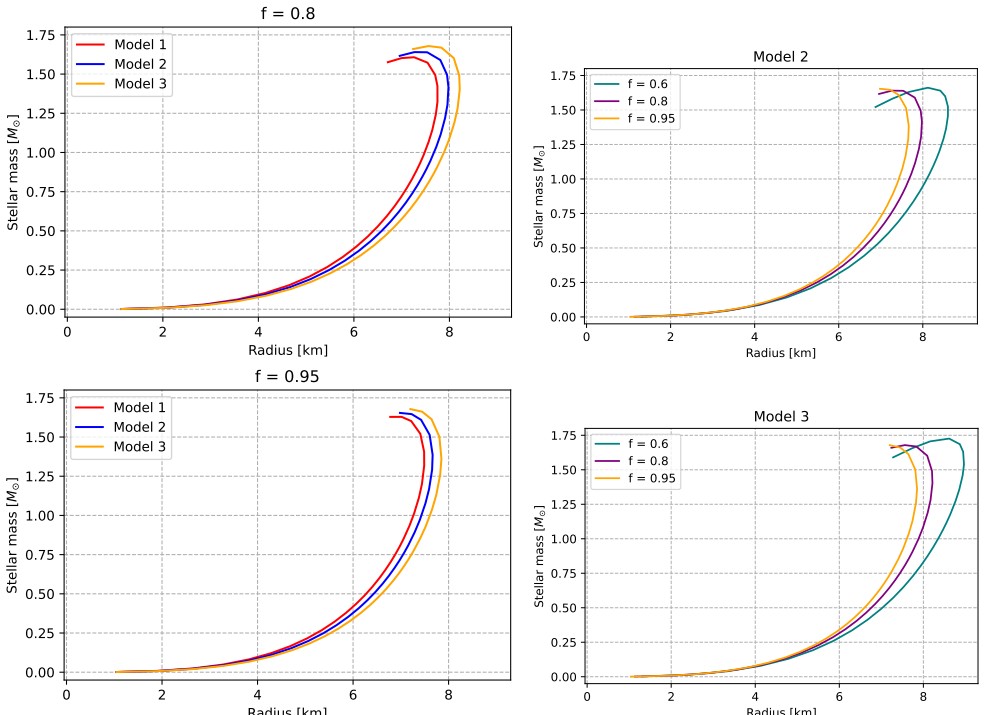

**Figure 1.** Mass-to-radius relationships (mass in solar masses and radius in km) for the exotic configurations considered in this work. In the first column, we vary the DE model for a given value of the factor $f$, while, in the second column we vary $f$ for a given DE model. See text for more details.

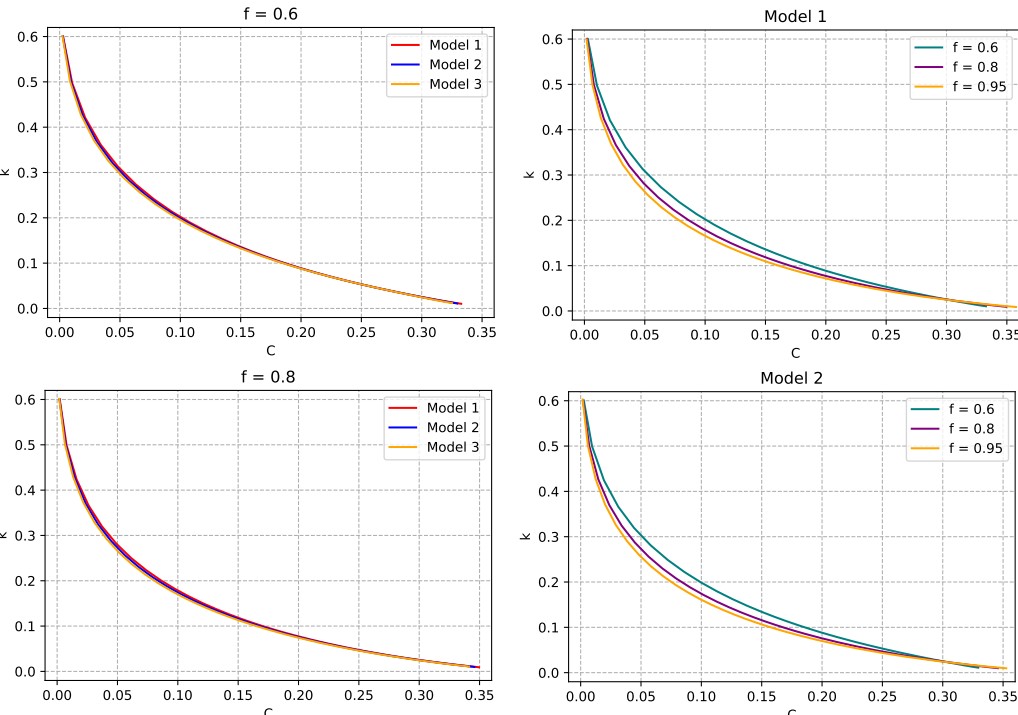

**Figure 2.** *Cont.*

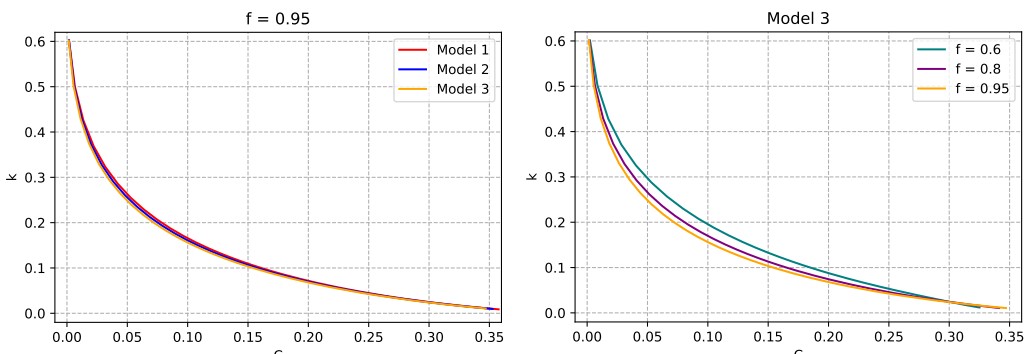

**Figure 2.** Tidal Love number, *k*, as a function of the factor of compactness, $C = M/R$. As in the previous figure, in the first column, we vary the DE model for a given value of the factor *f*, while in the second column, we vary *f* for a given DE model. See text for more details.

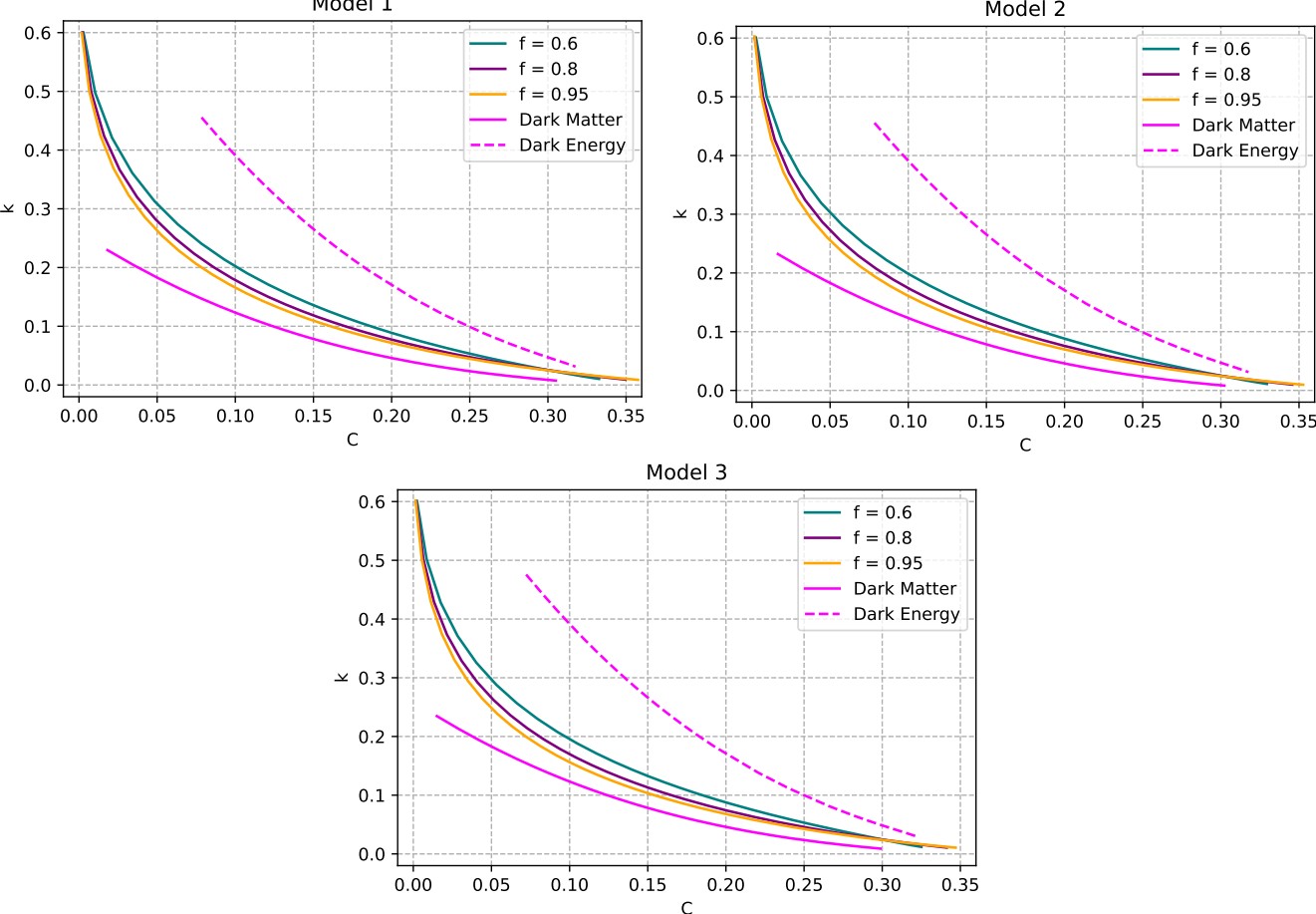

**Figure 3.** Tidal Love number, *k*, as a function of the factor of compactness, $C = M/R$. We fix the DE model (model 1), and we vary the factor $f = 0.6, 0.8, 0.95$. For comparison reasons, the predictions corresponding to the single-fluid objects (made of dark matter only or dark energy only) are shown as well. Notice that the dashed curves do not extend to zero compactness, the only reason being that it was difficult to generate it. The omitted portions, however, are not vital to the conclusions of this work.

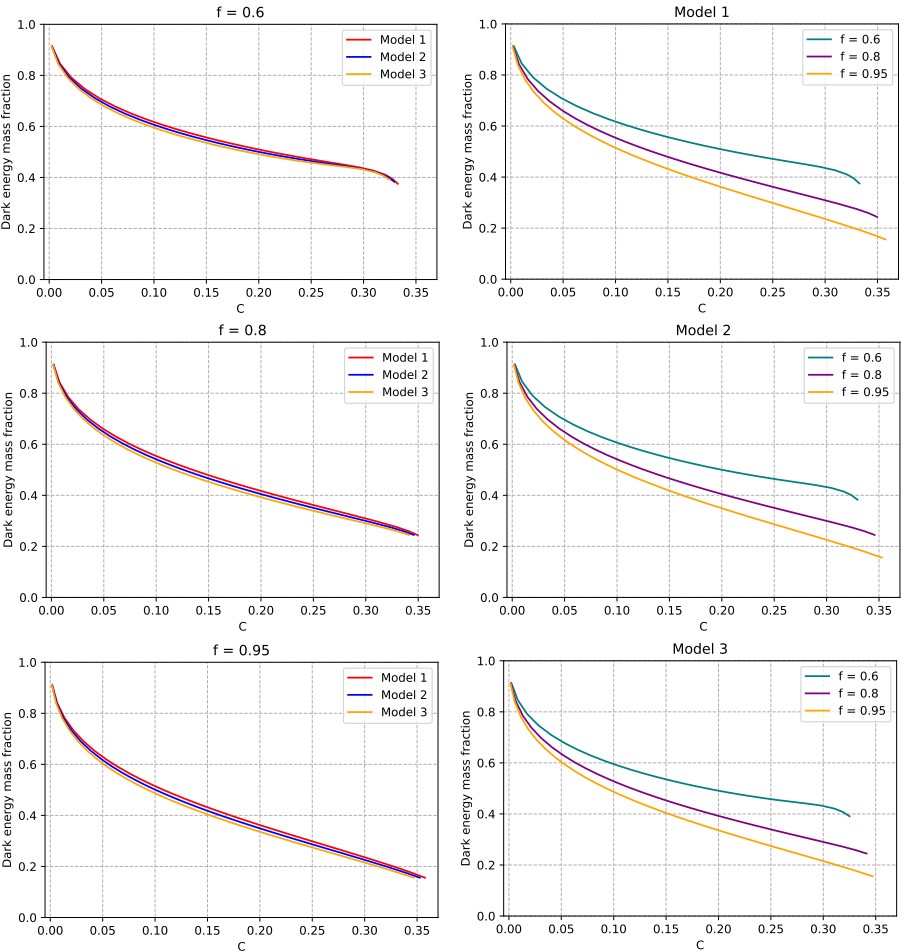

**Figure 4.** Dark energy mass fraction as a function of the factor of compactness. As in the first two figures, in the first column, we vary the DE model for a given value of the factor $f$, while in the second column, we vary $f$ for a given DE model. See text for more details.

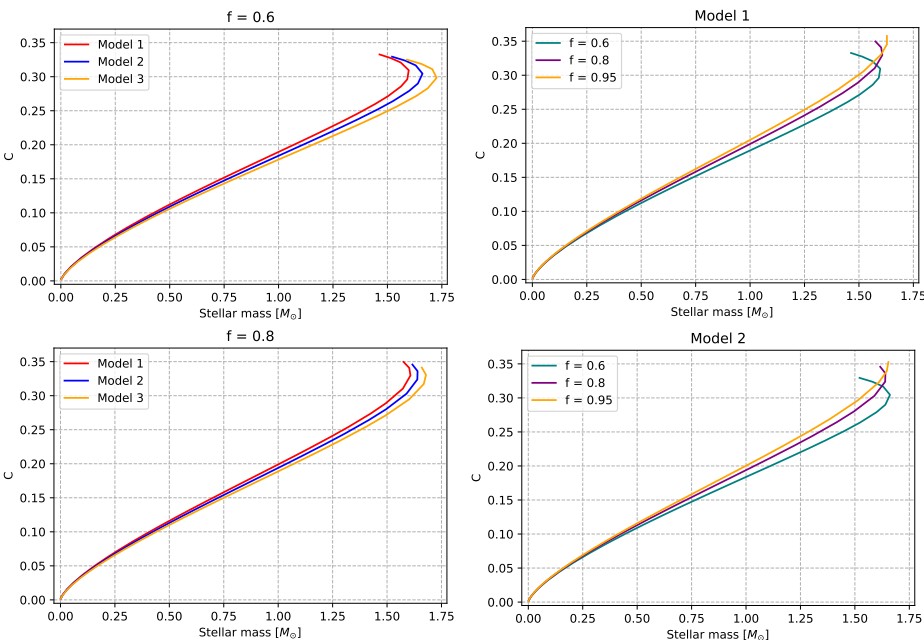

**Figure 5.** *Cont.*

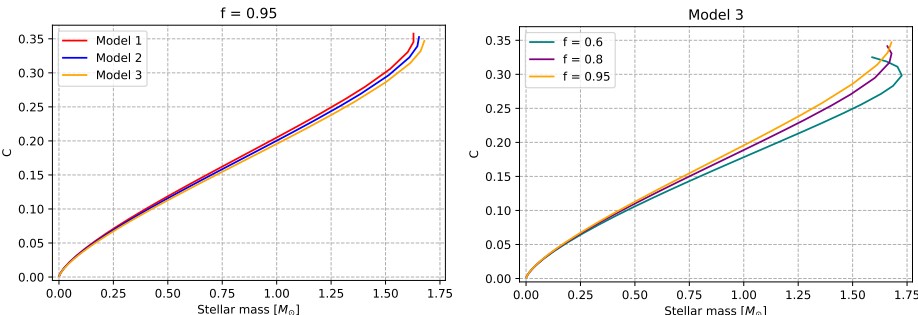

**Figure 5.** Factor of compactness vs. stellar mass (in solar masses). As in the previous figure, in the first column, we vary the DE model for a given value of the factor $f$, while in the second column, we vary $f$ for a given DE model. See text for more details.

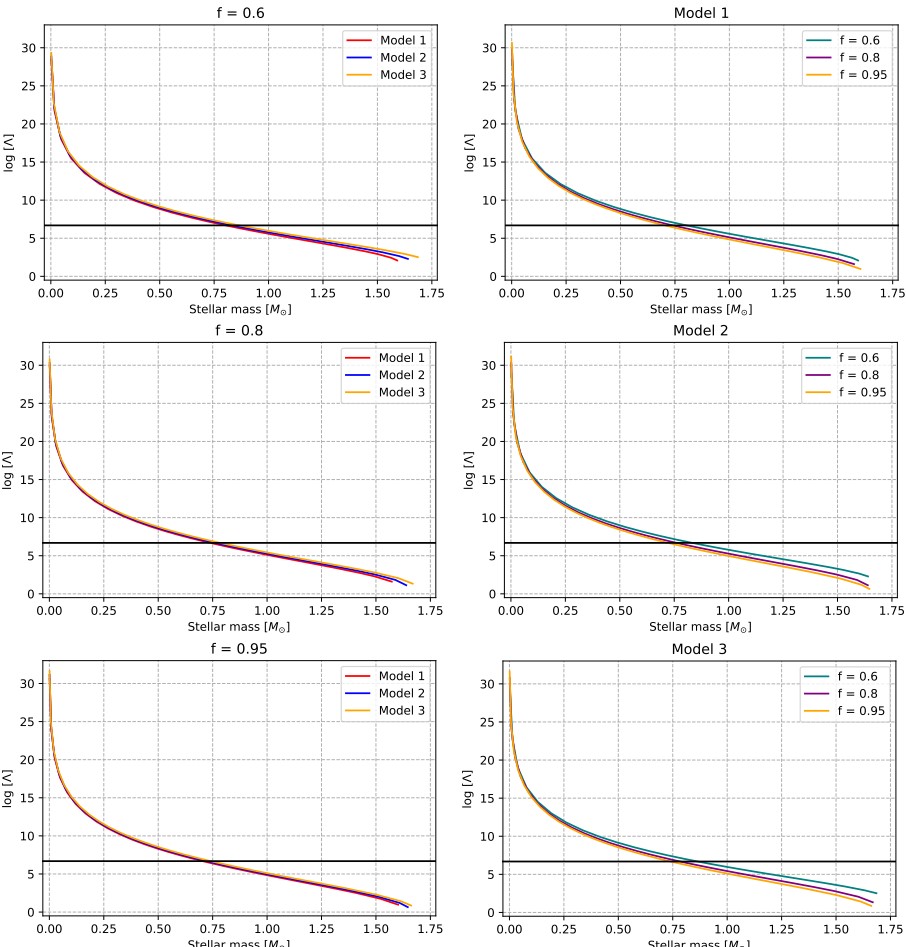

**Figure 6.** Natural logarithm (base e) of dimensionless deformability as a function of stellar mass (in solar masses). The horizontal line indicates the bound from the GW170817 [67] $\Lambda_{1.4} \leq 800$. As in the previous figure, in the first column, we vary the DE model for a given value of the factor $f$, while in the second column, we vary $f$ for a given DE model. See text for more details.

   In Figure 1, the M-R profiles are shown. In all six cases, the stellar radius reaches a maximum value first, and then the stellar mass also reaches its own highest value. We can notice that, for the same central pressure ratio, the most massive stars are those of model 3, while the least massive are those of model 1. Within the same models, the masses are similar, but in model 1, we can notice that they are more massive for $f = 0.95$, while in model 3, they are for $f = 0.6$. For a given value of $f$ (column on the left), the difference between different DE models is more important for lower values of $f$, whereas, as $f \to 1$,

the profiles lie one very close to another. On the contrary, for a given DE model (column on the right), there is always a considerable difference from one value of $f$ to another. Although the models considered in this work are not proposed to explain any observations of pulsars, we remark in passing that the maximum masses found here are notably lower than the largest measured pulsar masses (such as J1614-2230, mass at $1.908 \pm 0.016$ solar masses), see for instance [106].

In Figure 2, we display $k$ versus $C$ fixing the $f$ factor (column on the left) or fixing the DE model (column on the right). In all six cases, the curves qualitatively exhibit the same behavior. Namely, they monotonically decrease tending to zero as $C \to 0.5$, which is the black hole limit as already mentioned previously. For a given $f$ and varying the DE model, we observe no significant difference from one case to another. Contrary to that, the difference from one case to another is comparable to all three cases when we fix the DE model and vary $f$. Moreover, the curves are shifted downwards as we increase $f$. To compare with previous results on neutron stars or quark stars, it was found in [64] that the figures $k$ versus $C$ for quark stars are quite different in comparison to hadronic EoSs. To be more precise, in the case of NSs, the curves exhibit a pronounced maximum, as can be seen, for instance, in Figure 4 of [61] or Figure 7 of [64], and the maximum value lies between 0.1 and 0.14. On the contrary, in the case of quark stars, the tidal Love number may be as high as 0.8. Our results show that with a tidal Love number as high as 0.6, and without a pronounced maximum, the configurations studied here look more like quark stars than NSs.

In Figure 3a, the comparison is made between single-fluid objects, made of dark matter only or dark energy only, and two-fluid objects made of both dark components investigated here. The curve corresponding to pure DM objects (polytrope with $n = 1$) is the one shown in Figure 1 of [61,62]. The two-fluid case lies in between and approaches the pure DM curve as we increase $f$. The tidal Love number corresponding to the pure dark matter case is the lowest for a given compactness, whereas the curves corresponding to the two-fluid case are qualitatively similar to the curves corresponding to pure dark energy case. The curves corresponding to pure DE (dashed magenta) do not extend to zero compactness, the only reason being that it was difficult to generate it. The omitted portions, however, are not vital to the conclusions of this work, since a very low factor of compactness implies a non-relativistic object, whereas here we are mostly interested in compact configurations.

Next, in Figure 4, we show the DE mass fraction, $M_{DE}/M$, as a function of the factor of compactness. As in previous figures, we vary the model fixing $f$ in the column on the left, and we vary the $f$ factor for a given model in the column on the right. In all six cases, we observe that the DE mass fraction monotonically decreases with $C$, and that the factor of compactness always remains lower than $4/9$ in agreement with the Buchdahl limit [107]. The objects are less compact when they are DE-dominated, and more compact when they are dark matter dominated. Furthermore, for a given $f$, there is no significant difference from one model to another, whereas within the same DE model, the DE mass fraction decreases with $f$ for a given value of $C$.

The factor of compactness versus the stellar mass is displayed in Figure 5. In all six cases, the curves exhibit the same behavior qualitatively, where $C$ increases with $M$. The highest value of the mass and the maximum value of the compactness are the same as the ones shown in Figures 1 and 4, respectively.

Finally, the dimensionless deformability versus stellar mass (semilogarithmic plot) is shown in Figure 6. $\Lambda$ decreases with $M$ monotonically. In all six panels, the three curves lie very close to another, and some difference is observed for stellar masses at around 0.75 solar masses. The difference from one model to another for a given $f$ (column on the right) is somewhat more significant compared to the panels in the column on the left, where we fix the $f$ factor and we vary the DE model. In all six cases, the dimensionless deformability is in agreement with the bound $\Lambda_{1.4} \leq 800$ coming from GW170817 [67].

## 4. Summary and Conclusions

To summarize our work, in the present article, we studied exotic, electrically neutral, spherical configurations made of both dark energy and self-interacting dark matter. We employed the two-fluid formalism within Einstein's general relativity without a cosmological constant. We have adopted the extended Chaplygin gas for dark energy, and a polytopic EoS for condensed dark matter. The numerical values of the parameters of the models correspond to realistic, well behaved solutions describing objects with masses and radii similar to those of neutron stars and quark stars. We numerically integrated the structure (TOV) equations describing hydrostatic equilibrium to compute the mass, the radius, and the factor of compactness of the exotic configurations. Moreover, we numerically integrated the Riccati equation for metric perturbations in binaries to compute the tidal Love numbers as well as the dimensionless deformabilities. Our main results are displayed in the six figures of six panels each, where we vary both the dark energy model (models 1, 2, and 3) and the dimensionless factor $f$, as can be seen in the text for more details. Finally, we made a direct comparison between the two-fluid objects versus single-fluid spherical configurations made of dark matter only or dark energy only. For some works that consider distinguishing between different binaries, see, e.g., [108–111].

**Author Contributions:** Conceptualization: G. P., Methodology: G. P., Formal analysis: G. P., Investigation: C. S. and G. P., Writing original draft preparation: C. S., Software: C. S., Writing—review and editing: C. S. and G. P., Visualization: C. S. and G. P., Supervision: G. P. All authors have read and agreed to the published version of the manuscript.

**Funding:** The author C. S. acknowledges financial support from Universidad de la Frontera.

**Data Availability Statement:** This is a theoretical work and so no data have been reported here.

**Acknowledgments:** We are grateful to the reviewers for useful comments and suggestions.

**Conflicts of Interest:** The authors declare no conflict of interest.

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
