# Peer review of "On Exotic Objects Made of Dark Energy and Dark Matter: Mass-to-Radius Profiles and Tidal Love Numbers"

_galaxies, doi:10.3390/galaxies11050101_

Round 1

Reviewer 1 Report

This paper computes mass-radius relations and tidal deformabilities for stars consisting of dark matter and dark energy, using specific perfect fluid models for both. This is a straightforward calculation with standard methods, and there is only a cursory analysis of the results, with no explicit discussion of the prospects for gravitational wave constraints, even though this is given as a motivation. I think that for publication in Galaxies detailed data analysis calculations are not necessary, but a more detailed discussion of these prospects, with references to the literature, would be necessary, as mentioned below.

Additionally, there appears to be an error in the calculation: Since the energy density at the surface is nonzero in these stars, the Love numbers computed need a correction—see, e.g., Eq. (15) in doi:10.1103/PhysRevD.81.123016.

The statement in L232-4 that "Our results suggest that with the help of current and/or future gravitational wave detectors we may discriminate between different classes of stars (black holes, neutron stars/quark stars, exotic/dark stars)." needs to be substantiated with a comparison of the differences in tidal deformability versus mass computed here with those for, e.g., neutron stars, and a discussion of the observational prospects for distinguishing these (where one can draw analogies with constraints on the neutron star EOS, e.g., doi:10.1103/PhysRevD.107.124006 for O4 and arXiv:2205.01182 for 3G detectors). It is likely also appropriate to cite other works that consider distinguishing between binary neutron stars, neutron-star black hole systems, and/or binary black holes, e.g., dois: 10.3847/2041-8213/ab86bc, 10.1103/PhysRevD.101.103008, 10.1103/PhysRevD.102.023025, and 10.1103/PhysRevD.105.064063.

It would also be appropriate to compare the tidal deformabilities computed here with the constraints from GW170817.

I also have some other comments that should be addressed before this is considered for publication.

* Sec. 1:

L20: This should also cite some more recent observations of galaxy rotation curves and/or a review discussing this evidence for dark matter.

L25: Similarly, this should cite some more recent cosmological constraints from supernova observations. If the current citations are kept, it should be clarified that they are historical.

L27: This should clarify that this is only true for sufficiently massive stars—brown dwarfs.

L31: This should presumably mention boson stars and give the citations given in L41. It should also be possible to rewrite things so that there’s no such duplication. The citations for boson stars also should include the standard Living Review (doi:10.1007/s41114-023-00043-4), and at least one or two citations of papers about numerical simulations of binary boson stars (e.g., arXiv:2306.17265 and possibly also doi:10.1088/1361-6382/acc2a8, though there are others).

L42-4: This sentence conveys the same information as the sentence in L34-5, so only one of those sentences needs to be kept.

L46: \alpha isn't introduced explicitly

L52: The statement that the tidal forces come from strong gravitational interactions seems odd—all true gravitational forces are tidal, not just strong ones.

L53: Since you mention Love, you should presumably cite his work here, as you do in L110.

L62-5: You should cite some of the papers giving the LIGO-Virgo observational results (at least the GW170817 paper and the latest GWTC-3 catalogue paper; see https://www.ligo.caltech.edu/page/detection-companion-papers for all the LVK detection papers). This should also clarify that KAGRA hasn’t detected any gravitational wave signals yet, and that LISA is still in the future. It’s probably also appropriate to mention the third-generation ground-based detectors Cosmic Explorer and Einstein Telescope, since these should be observing at the same time as LISA.

L65-8: This should clarify that space-based detectors are not expected to constrain the NS EOS, and one cannot use black hole-black hole mergers to constrain this (except a bit indirectly if one observes a low-mass black hole and assumes that the minimum astrophysical black hole mass is above the maximum neutron star mass, which I presume is not what is intended here). Additionally, the tidal effects used to constrain the EOS are only prominent in the late inspiral, not in the early low-frequency part, as stated. Moreover, one would also be able to use the high-frequency post-merger part of the signal to constrain the EOS, if this was observed.

This should also cite some papers describing the constraints on the EOS one obtains from GW observations (e.g., the LIGO-Virgo observational result for GW170817 in doi:10.1103/PhysRevLett.121.161101, but likely also some more recent papers, e.g., doi:10.1103/PhysRevD.101.123007, but there are many others). You should probably also cite some papers describing constraints on boson stars with GW observations, e.g., dois:10.1103/PhysRevD.104.084056 and 10.1103/PhysRevD.108.023021. In fact, it might be more useful to discuss this possibility and similar possibilities for constraining exotic compact objects, as opposed to the possibility of constraining the EOS of neutron stars (though it's fine if you want to mention this, as well, since it is currently a topic of considerable interest).

Finally, the citation for LISA is not to a paper describing the instrument. There’s also a bibliography entry for the LISA website ([56]) that I can’t find cited in the text, but you probably want to cite arXiv:1702.00786 instead, and can also cite [57] in addition, if you want to.

* Sec. 2.2:

In Eq. (20), c_s is not defined. Additionally, the statement in L120-1 is unclear. Is what is meant that if one uses Eqs. (19) and (20) as written (with some appropriate definition of c_s in this multi-fluid case) then one obtains the same results as if one makes the substitution given in Eq. (21) in Eq. (20)? If so, this can be said more explicitly.

L122: Here (1 - 2C) should presumably be (1 - 2C)^2 [cf. Eq. (7) in [52]], though since y_R also has dependence on C, this should be clarified.

You need to define \Lambda somewhere, probably in this subsection.

* Sec. 3.2:

Since you mention in Sec. 3.1 that emulating neutron star masses is a motivation for the specific parameters considered, it's probably worth remarking explicitly on the fact that the maximum masses found here are notably smaller than the largest measured pulsar masses (e.g., J1614−2230 at 1.908 \pm 0.016 Msun, see doi:10.3847/1538-4365/aab5b0). Since the models given here are presumably not proposed to explain any observations of pulsars, this is not a big problem, but it's good to point this out explicitly.

In Fig. 3, is there a reason why the dark energy curves do not extend to 0 compactness?

* Minor:

L41: Presumably [42] and [43] should be cited chronologically (so in the opposite order) 

L143-4 and Table 1: The units should be set in Roman

I noticed a number of typos, some of which are given below, but the English is overall fine.

L72: "numebrs"

L76: "love" -> "Love"

L78: The first "fourth" should be "third"

L89: "differentiantion"

L127: "elctromagnetic"

L138: "rice" -> "rise"

[68, 69]: The initial should be "A."

Author Response

Dear Editor,

Please find our response in a separate file.

Sincerely,

Grigoris Panotopoulos

Reviewer 2 Report

This is an interesting investigations of bulk properties of exotic, electrically neutral, spherical and selfgravitating configurations made of separately conserved dark energy and self-interacting dark matter. For the former, an extended Chaplygin gas EoS, for the latter a polytopic EoS was employed. Numerical analysis of the correspondingTolman-Oppenheimer-Volkoff equations yield realistic, well behaved solutions describing objects similar to those of neutron stars and quark stars if the EoS parameters are chosen appropriately. In addition, the authors compute on these solutions the bulk perturbation measures, relevant to the characterisation of gravitational-wave emitting binaries of such objects, such as the tidal Love number and the dimensionless deformability Lambda.

I believe that the analysis presented here is interesting and useful. As far as I can see, it is also correct. Therefore, I recommend the publication of this manuscript in Galaxies after the following minor points are mitigated by the authors.

1) l. 56: be also -> also be

2) after Eq. (6) introduce definitions of total pressure and energy density immediately, otherwise readers may get confused about meaning of rho and p in Eqs. (3) - (7)

3) l. 109: Tidal -> The tidal

4) throughout: introduce a dot after an equation and use kommas to separate equations

5) l. 127: elctromagnetic -> electromagnetic

6) citing the fuzzy-dark matter paradigm in line 137, [72-75], one may also be interested in cosmological implications, see https://www.mdpi.com/2218-1997/7/6/198 . I'd also direct the attention of the reader to some of the pioneering papers: 

Ji, S.; Sin, S.J. Late-time Phase transition and the Galactic halo as a Bose Liquid: (II) the Effect of Visible Matter. Phys. Rev. D 199450

Hui, L.; Ostriker, J.P.; Tremaine, S.; Witten, E. Ultralight scalars as cosmological dark matter. Phys. Rev. D 201795, 043541. 

Schive, H.Y.; Chiueh, T.; Broadhurst, T. Cosmic Structure as the Quantum Interference of a Coherent Dark Wave. Nat. Phys. 201410, 496–499.

Bernal, T.; Fernández-Hernández, L.M.; Matos, T.; Rodríguez-Meza, M.A. Rotation curves of high-resolution LSB and SPARC galaxies with fuzzy and multistate (ultralight boson) scalar field dark matter. Mon. Not. R. Astron. Soc. 2018475, 1447–1468.

6) l. 168: diformability, Lambda -> deformability Lambda; Lambda is never defined in the present work which should be done!

English is sufficiently good, but can be improved. 

Author Response

(The authors gave the same response as above.)

Round 2

Reviewer 1 Report

I thank the authors for their edits and reply, which has addressed a number of my comments. However, there are still two major issues as well as a number of smaller issues that need to be corrected before this is published:

The correction due to the finite surface density is needed for these results, since it is present even in cases where the speed of sound is well-behaved and nonzero inside the star. See Eq. (99) in Damour and Nagar [62], which shows that there is a contribution that depends on d\rho/dr, and thus there will be a delta distribution contribution from this at the surface in cases where the surface density is finite.

Additionally, the single sentence just giving references about distinguishing different types of binaries with GW observations added at the end of the conclusions is not sufficient to let the reader determine how easy it would be to distinguish the models presented here with current or future GW detectors. Thus, this discussion needs to be expanded to give such information before this is published. Additionally, Refs. [106, 107] consider constraining the EOS, not distinguishing between different types of binaries.

* Sec. 1

The more recent citations for supernova cosmology constraints should cite some papers that perform the observations, at least doi:10.3847/1538-4357/ac8e04, and possibly also doi:10.3847/2041-8213/ab04fa. One might replace one or more of the current citations with these citations, but it is also fine if you just cite all of them.

L31: The comment about “in massive stars” is unclear. This was presumably added in response to my previous comment “L27: This should clarify that this is only true for sufficiently massive stars—brown dwarfs.” However, this comment was unfortunately incomplete (apologies!) and meant to convey that one only gets neutron stars or strange quark stars (if the latter exist) as the final state of sufficiently massive stars, since less massive stars lead to white dwarfs, which are considerably less relativistic. I thus think that the mention of sufficiently massive stars should be added in the sentence in L28-9. I had meant to remove the comment about brown dwarfs, since I realized that these are not usually considered to be proper stars.

L34: [28, 29] are not references about boson stars, but rather Bose-Einstein condensate dark matter, so they do not need to be cited here. Apologies for not realizing this in my previous report. I would also recommend citing the two reviews separately, not just in the middle of the block of citations, and mentioning that [27] is the up-to-date review. Similarly, the two papers about numerical simulations of binary boson stars [32, 33] should be cited separately instead of just in the block of citations about the proposal of boson stars.

L43-6: As I mentioned in my previous report, the “Similarly, given … just might exist [45–48].” sentence repeats almost verbatim the contents of L36-8 “Similarly, there … might exist as well [45–48].” Thus only one of these should be kept.

L63-4: “Due to the collision of compact objects between neutron stars or between black holes in binary systems” is oddly worded. This could be written more clearly as something like: “Through observations of gravitational waves from mergers of binaries containing neutron stars and/or black holes”; apologies for not mentioning this in my previous report. The end of this phrase would also be the appropriate place to cite the LIGO-Virgo observation papers I mentioned in my previous report. You already cite the GW170817 discovery paper [71] elsewhere, but it also seems appropriate to cite that here, as well as the latest catalogue paper (arXiv:2111.03606).

The discussion in L65-73 can still be clarified, e.g., for L65-7, writing something like “”…thanks to the gravitational wave (GW) detectors LIGO [65] and Virgo [66]. KAGRA [67] will also contribute to future such observations, and there are future ground-based GW detectors planned (Einstein Telescope and Cosmic Explorer) that will provide very sensitive measurements of such binaries. There are also planned low-frequency space-based GW detectors such as LISA [68-70].” Here you might also mention LIGO-India after KAGRA, since ET and CE are further in the future—apologies for not mentioning this in my previous report. Also, there should be citations for ET and CE. You could also remove the mention of LISA entirely, since it won’t be able to probe the tidal deformability of objects with the masses you consider here. If you keep the mention of LISA, I recommend removing the citation to [68], since it seems odd that you cite the webpage for LISA but not for any of the other detectors.

You say in the reply that “Ground-based detectors are not expected to constrain the NS EoS” and in L68-9 that “future space-based GW detectors may be able to constrain the nuclear EoS,” which are incorrect—I am not aware of any studies that indicate that this will be the case. This should be corrected. (In L70-1 you correctly say that “space-based detectors are not expected to constrain the neutron star EoS.”)

Additionally, in L69 you can cut “the frequencies of” since while the constraints on the EOS mostly come from the signal’s phasing, there is information in the amplitude, as well. In L69-70, you can just cut “black hole-black hole” and then won’t need the caveat about this in L71-2.

In L73, Refs. [74-75] deal with constraints on boson stars from putative binary black hole signals, not the nuclear EOS. It’s fine to cite these papers here, but the text needs to be clear about their contents.

* Sec. 2.2:

Eq. (23) can be removed, or at least combined with Eq. (24).

Both sides of Eq. (25) give -2\rho’(r)/\nu’(r), using Eqs. (4-8) and the definition of c_s^2 below Eq. (20), so it is no surprise that both of them give the same results numerically. This should be clarified.

L145: It is probably clearer to refer to something like "the overall factor of (1 - 2C)^2 in k" instead of writing k \propto (1 - 2C)^2, since this implies that this gives the only dependence of k on C, while there is other explicit dependence, as well as implicit dependence in y_R.

* Sec. 3.2:

The added text in L203-6 should recall the motivation given in the previous subsection of modeling objects with properties similar to neutron stars, to explain why you are comparing with pulsar masses. It should also quote the well-measured mass of J1614−2230 from [100] explicitly.

It seems odd not to let the dark energy curves in Fig. 3 extend to zero compactness, and in particular that the curve for Model 3 ends at a larger compactness than for Models 1 and 2. I thus would recommend rerunning to generate the missing portions of these curves. However, I agree that the omitted portions of the curves are not vital to the conclusions of this paper, so if it is not easy to generate them, for some reason, then the present note in the text is sufficient (though an additional short note in the caption about this might also be appropriate).

In Fig. 6, it’s probably worth noting explicitly that the logarithm is (presumably) base 10, not base e (which is also denoted by just “log” in some contexts). This could either be done by replacing log with log_{10} in the axis labels or with a note in the caption. Apologies for not thinking to point this out in my previous report.

L246: It is probably a good idea to say explicitly that this bound comes from GW170817.

There are still some typos, so this should be proofread carefully. I list a few of the ones I noticed:

L144: "numers"

L192: “Lover”

Fig. 3 caption: “signle”

Author Response

Dear Editor,

We have included our response as a separate file, please see the attachment.

Sincerely,

Grigoris Panotopoulos

Round 3

Reviewer 1 Report

Thanks for fixing the tidal deformability computation to include the effect of the density discontinuity at the surface. However, the text currently doesn’t mention that you include this, instead just mentioning (L133-5) that this correction isn’t necessary inside the star. This needs to mention that you included the correction and I think it should give the explicit expression used, since all the other equations are given explicitly.

Most of my other comments were also addressed quite satisfactorily, though I still have some small additional comments about some of them, given below. However, my major comment about needing to discuss how easy it would be to distinguish these objects from other compact objects using GW observations was not addressed. If this is not added (at least discussing the differences in tidal deformability versus mass and the expected observational errors on these, if not considering population-level statements), then the statements at the end of the abstract (“We also briefly comment on how to distinguish between different classes of objects analyzing the gravity waves emitted from the collision of compact objects in binaries.”) and at the end of the conclusions [“Our results suggest that with the help of current and/or future GW detectors we may discriminate between different classes of stars (black holes, neutron stars/quark stars, exotic/dark stars).”] should be removed or significantly revised. However, I think that it would be more useful to the reader to discuss the observational prospects for distinguishing these objects.

L36: I think that the citations for the numerical simulations of binary boson stars (currently [71,72]) can go here, and one can refer to the reviews more clearly, e.g., something like “stars [23-28] (see, e.g., [71,72] for  recent progress in numerical simulations of boson star binaries). See [29] for a classic review of boson stars and [30] for a very recent one.” If you want to refer to the numerical simulations of boson star binaries elsewhere, that’s fine, but the current mention in L73 seems very out of place and just part of a laundry list of citations that doesn’t really help the reader.

L69: [67] isn’t the standard CE citation, which would instead be arXiv:2109.09882.

L72-3: The statement “for G3 detectors see the latest catalogue paper” (then citing the LVK GWTC-3 paper) is very unclear. In particular, the mention of “G3 detectors” makes it sound like this is referring to third generation (3G) detectors such as Einstein Telescope and Cosmic Explorer. I suggest citing the catalogue paper in L67 after “thanks to the GW detectors LIGO [62] and Virgo [63]” with something like “; see [70] for their latest catalogue paper.” You might also cite the GW170817 paper here, as I suggested in my previous report. Since I suggest moving the binary boson star simulation references earlier, one can just start the sentence in L72 with “For constraints on the nuclear equation-of-state” which will make much more sense.

L190: “super massive pulsars” is not standard terminology. I suggest just something like “most massive pulsars.”

L216: This quotes the original mass measurement of J1614-2230, not the updated one (1.908 \pm 0.016 Msun) from [106].

L226: It’s probably also appropriate to refer to Hinderer et al., doi:10.1103/PhysRevD.81.123016 (which was before [61]) for the tidal deformability of quark stars. They give a nice comparison in Fig. 1, though you don’t have to refer explicitly to the figure if you don’t want to.

For Fig. 6, I think that the base 10 logarithm would be easier to interpret, but will not insist on this change.

The English is generally fine, though there are still a few typos, e.g.,

L190: "discuassion"

L227 & 230: "pronounce" should be "pronounced"

Author Response

Dear Editor,

We have submitted a new revised version of our manuscript.

Our response can be seen in a separate file, please see attachment.

Sincerely,

Grigoris Panotopoulos
